# Experiences and Opinions of Physiotherapists, Children, Families, and Teachers About School-Based Physiotherapy-Led Interventions: A Metasynthesis of Qualitative Studies

**DOI:** 10.3390/healthcare13222859

**Published:** 2025-11-11

**Authors:** Gonzalo Granero-Heredia, Nuria Sánchez-Labraca, José Granero-Molina, Rubén Fernández-García, Laura Helena Antequera-Raynal, Susana Núñez-Nagy

**Affiliations:** 1Nursing, Physiotherapy and Medicine Department, University of Almería, 04120 Almería, Spain; gonzalograneroheredia@gmail.com (G.G.-H.); msl397@ual.es (N.S.-L.); rubenfer@ual.es (R.F.-G.); lar855@ual.es (L.H.A.-R.); 2Faculty of Health Sciences, Universidad Autónoma de Chile, Santiago 7500000, Chile; 3Vithas Hospital, 04120 Almería, Spain; 4Department of Nursing and Physiotherapy, Faculty of Medicine and Health Sciences, University of Alcalá de Henares, 28805 Alcalá de Henares, Spain

**Keywords:** physiotherapy, school health, special needs, physiotherapy services, qualitative research

## Abstract

**Background/Objectives**: Physiotherapy is a clinical discipline that provides services in healthcare centres, the community, at home, and in school. School Physiotherapy (SP) focuses its practice on children with disabilities or special motor needs during the school period. SP addresses psychomotricity, ergonomics, and posturology issues in order to improve health and academic performance. The objective of this study was to identify qualitative evidence on the experience and opinions of physiotherapists, children, parents, and teachers about school-based physiotherapy-led interventions. **Methods**: Metasynthesis of qualitative studies. The search included articles published between 2015 and 2025 on the PubMed, WOS, SCOPUS, and CINAHL databases. **Results**: The review included 15 selected studies. Three main themes emerged from this analysis: (1) Clinical and school physiotherapy: separated by a fine line. (2) Ensuring healthcare for children with specific conditions in schools. (3) The challenge of incorporating SP in educational settings. **Conclusions**: Physiotherapists, children, parents, and teachers perceive significant advantages in implementing SP in educational settings. Participants advocate for the development of collaborative goals and the inclusion of physiotherapists within educational teams. Understanding the experiences of physiotherapists, children, parents, and teachers may help improve SP in practice.

## 1. Introduction

Physiotherapy is an autonomous profession whose objective is to prevent, develop, and maintain movement and functional abilities in patients and communities. According to the World Confederation for Physical Therapy, the practice of the profession varies depending on the healthcare system, facilities, regulations, and resources of each country [1]. Although physiotherapy is essentially a clinical discipline, carrying out the assessment, diagnosis, and treatment of patients, it currently forms part of integrated models that provide services in schools [2,3]. The interaction between healthcare and educational development has evolved in recent years [4]. Institutions seek cooperation between the educational, healthcare, and social sectors [5]. Schools provide a unique natural environment for physiotherapists, as they can promote physical activity and develop motor learning strategies or early intervention therapies [6,7]. School Physiotherapists (SPs) focus specifically on the care of children with special educational needs, developing physical rehabilitation programmes, physiotherapy assessment, guidance, and treatment of neurorehabilitation and motor problems [8]. In several countries, SP has already been established as an integrated service within the educational system [9,10,11]. By integrating theory, scientific evidence, and clinical knowledge, SP provides technical aids and curricular adaptations for children with disabilities, and motor, ergonomic, or postural problems [7,12,13]. Classroom teaching is not free from issues concerning psychomotricity, ergonomics, or posturology, many of which can be prevented and managed through SP services [14]. At the same time, SP can provide guidance on physical activity [15,16,17], the reduction in sedentary behaviour, musculoskeletal disorders [18], occupational health [14], postural health [19], and adapted teaching in special education settings [20]. The interaction between education and health can have a positive impact on the academic success and quality of life of students with disabilities [10,21]; however, it is necessary to document the strengths and weaknesses of SP services, the gap between ideal and current practice [22], and to evaluate the effectiveness of programmes [10]. Although some studies have reported discrepancies regarding SP interventions [23], most agree on their benefits for improving children’s motor and social skills [24]. Several studies point to a lack of evidence concerning therapeutic interventions, rehabilitation, or educational support within the classroom context [9,25]. Furthermore, a deeper understanding of the experiences of the participants involved in this process is required [26]. Qualitative methodologies have proved useful in researching the experiences of physiotherapists, families, teachers, or children regarding SP for cerebral palsy [9], the use of standing frames [27], aquatic therapy [28] or spinal health programmes [29]. In order to improve the quality of care in educational settings, there is a need for a synthesis of aggregated data in order to obtain a deeper understanding of the experiences, opinions, use, development, strengths, and weaknesses of SP services. The objective of this study was to identify qualitative evidence on the experiences/opinions of physiotherapists, children, parents, and teachers about school-based physiotherapy-led interventions.

## 2. Materials and Methods

### 2.1. Design

A systematic review and metasynthesis of qualitative studies was performed.

### 2.2. Search Methods

Bibliographic searches were conducted on the PubMed Central, WOS Core Collection, SCOPUS, and CINAHL complete databases for qualitative studies in English and Spanish, published between January 2015 and October 2025 (Table 1). The Enhancing Transparency in Reporting the Synthesis of Qualitative Research declaration was used in this study (Appendix A) [30]. The SPIDER method was used to formulate the research question: Sample (physiotherapists, children, parents, teachers), Phenomenon of Interest (any school-based physiotherapy-led interventions); Design (descriptive study, phenomenological study, …); Evaluation (experiences, opinions); Research (qualitative, mixed methods) [31].

### 2.3. Inclusion and Exclusion Criteria

Inclusion criteria: primary studies on the experiences and opinions of physiotherapists, parents, teachers, and children about any school-based physiotherapy-led interventions. Qualitative research papers or mixed methodology were included. Exclusion criteria: non-primary articles, editorials, or opinion pieces.

### 2.4. Search Results

A 5-stage selection process was performed (January/February 2025) by three researchers (GGH, LHAR and SNN A total of 967 studies were identified; however, only fifteen studies were included in this review (Figure 1).

### 2.5. Data Extraction

Two authors (GGH, NSL) independently performed data extraction and reached a consensus. Disputes were discussed with a third researcher (SNN). Articles selected in each phase were approved by ≥2/3 of the researchers. (Figure 1).

### 2.6. Quality Assessment

Each primary study was assessed using the Joanna Briggs Institute’s Qualitative Assessment Rating Instrument (QARI) [32]. The included articles were considered to be of high quality with respect to objectives, design, analysis, and results, providing useful knowledge on the topic (Table 2). This review included all studies that met ≥ 70% of QARI quality criteria; no studies were excluded after quality assessment.

### 2.7. Data Synthesis and Analysis

The included studies were analysed thematically. The synthesis was undertaken by (G.G.-H. and N.S.-L.) and verified by (S.N.-N.). One independent reviewer with expertise in SP and qualitative research verified the results. The thematic synthesis included three stages [39] (Table 3). Characteristics of selected studies can be seen in Table 4.

## 3. Results

The fifteen qualitative studies comprised a total sample of 560 participants, 179 (31.9%) children, 65 (11.6%) parents/relatives, 124 (22.1%) teachers, 175 (31.2%) physiotherapists, and 17 (3%) unspecified participants. Regarding their origin, there were four studies from South Africa, three from the USA, three from Spain, two from Australia, one from Norway, and one from Israel, the UK, and unspecified countries. Thematic synthesis is an inductive process in which three themes and twelve sub-themes emerge (Table 5).

After the thematic synthesis, the contributions of the studies to the final analytical themes and subthemes were examined (Table 6). An example of the coding process, from quotations to codes, units of meaning, subthemes, and themes, has been included (Table 7).

### 3.1. Clinical and School Physiotherapy: Separated by a Fine Line

SP differs from clinical physiotherapy; SP services should represent a potential for educational improvement in children with disabilities. Although the whole multidisciplinary team shares the same objective, it is difficult to strike a balance between physical activity, therapy, and academic work. Educators fight for education and physiotherapists for physiotherapy, but it is a joint task where neither is more important than the other; both must coexist:

*Well, what’s the most important thing here? Is it the standing? Is it education? Is it this bit? Is it that bit?” … It’s a very fine balancing line … constant battle between therapy and education*.[27]

Physiotherapists point to a paradigm shift in SP, highlighting the difficulty of moving from a clinical model to an educational one:

*I’ve been a PT for 27 years but I’ve only been in the schools for about 14. Certainly, coming from a clinic model, my first couple of years in the schools I had a really hard time to figure out how to make goals that were school-based…*.[3]

The introduction of physiotherapy in schools involves living with feeling under pressure while not forgetting that the goal is to improve health and education. One therapist explained:

*We’re looking at supporting students in special education and benefiting from those services; we’re looking at accessing educational environments and students participating with their peers in motor activities*.[3]

There is a lack of interaction, knowledge sharing, and communication between the clinical and academic fields, and this is perceived as a threat to the advancement of SP.

#### 3.1.1. Lack of Specific Guidelines and Ambiguity of Professional Competencies

SP is necessary when disabilities and motor problems negatively affect a child’s academic performance. A health problem that interferes with academic performance may require clinical physiotherapy and/or SP. However, as one physiotherapist notes, school settings lack protocols for action or specific tasks for physiotherapists:

*I like that it gives nice clarity around the scope of work for a school-based physiotherapist*.[20]

The present study highlights the existence of distinct guidelines, competencies, functions, and registries. As one participant states, this situation confuses physiotherapists. There is a need to organise SP practice:

*Sometimes because of the wording, complexity and differences between countries, PTs might feel so overwhelmed*.[20]

Defining the professional competencies of SP poses a challenge. The ambiguity with related professions means that some essential functions that define SP are assumed by other professions. While regulations aim to standardise the functions of SP, their implementation reveals gaps in accessibility, availability, and professional accountability.

*Actually, it is everyone’s responsibility, so when everyone is responsible, no one takes responsibility*.[11]

#### 3.1.2. Lack of Support: Feeling Like Outsiders

School physiotherapists report a lack of support from department heads and guidance from educational coordinators. Physiotherapists must be familiar with the school, but they are sometimes forced to adapt by taking on non-clinical activities outside of their professional duties, such as supervising recess or covering shifts for other professionals. There are no job descriptions, positions, training, or specific documentation.

*Most of the focus [in our school] is on education and therapists are excluded from the decisions [physiotherapists not consulted to share their views] by the DoE. … To date we do not have job descriptions and are understaffed*.[26]

School physiotherapists may feel like outsiders in an educational system where the majority are teaching staff. Physiotherapists perceive insufficient training, fewer opportunities for professional development, and isolation.


*Physiotherapy is secondary in the education system, even within the health professions… everyone needs to talk, so they [the students] receive speech therapy… Everyone needs to write and hold a pencil, so who will teach them? An occupational therapist… and due to various misconceptions, the OT will also examine motor performance. So why bring in another clinician?*
[11]

#### 3.1.3. Possible Professional Intrusion

Some policymakers expressed concern about professional boundaries. SP may overlap with occupational therapists, developmental coaches, physical education teachers, or sports coaches. Sometimes these professionals could offer physiotherapy treatments for children with motor disabilities.

*There is a big problem with physiotherapy professional boundaries: where do they begin and end, who needs physiotherapy, who should be referred to a physiotherapy… I mean, it is unclear to the public but also to therapists, when should a physiotherapy or occupational therapist be consulted?…*.[11]

#### 3.1.4. Children-Centred, Evidence-Based

Beyond teachers’ opinions, school physiotherapists are trained to write measurable goals and develop evidence-based activities. As one physiotherapist with 20 years of experience in school states:

*And I don’t know that there is really one great way to write goals but making sure that they’re measurable and you can actually keep data on those goals*.[3]

SP should focus on the pupil, addressing their access and mobility issues during the school day. The goals are geared toward motor development, independence in movement, gait, balance, and locomotor patterns. As one physiotherapist put it:

*So, if it was a clickable link, if it was an interactive document, we could digitally find more of the information in it*.[20]

### 3.2. Ensuring Healthcare for Children with Specific Conditions in Schools

Children with specific conditions such as cerebral palsy, spinal cord problems, anatomical malformations, or recurring pain require physiotherapy support during their daily activities at school.

#### 3.2.1. Care for Children with Cerebral Palsy

Children with cerebral palsy attend specialised schools that offer modified curricula along with therapy services. Their experiences with exercise are positive, improving their aerobic capacity, physical endurance, well-being, and participation in school and leisure activities. School physiotherapists state that they perceive improvements in motor skills, communication skills, and improved social relationships:

*She was walking longer on the treadmill and riding her bike faster for longer periods*.[17]

*I think that open friendliness, that it being a little bit social as well as exercise, was a great aspect of it. And so that would make it easier for them to join in*.[17]

Teachers and physiotherapists recognise the advantages of children with cerebral palsy participating in these programmes, which focus on addressing children’s academic and physical needs and responding to family expectations.

#### 3.2.2. Supervising Children Using Standing Frames

Using standing frames for children with specific pathologies requires knowledge of their handling and complications. While classroom staff may have general skills, prolonged use, agitation, or movement of children can cause injuries or burns. Using this technology requires training for students and a multidisciplinary team; this task must be carried out by the SP.

*If the children have got a lot of extraneous movement and they’re agitated, you can end up with friction burns … Sometimes it actually depends if they’ve got their second skin (dynamic lycra body suit) on, if they are tired … So you have to really know your children and know what mood they’re in as well*.[27]

#### 3.2.3. Prevention and Treatment of Back Problems/Pain

Knowing the consequences of poor posture helps children change habits and motivate them to improve their ergonomics. According to teachers, information provided through physiotherapy activities allows students to improve and prevent back pain. Children, parents, teachers, and principals emphasise the benefits of participating in physiotherapy workshops, such as risk awareness, pain reduction, and improved well-being. One child expressed the following:

*The workshops have helped me to change, I have truly felt that they have been useful. In my daily life, this is noticeable and little by little, the back ache that I had starts to go away*.[19]

For the children, the physiotherapy activities they applied most were those related to ergonomics, stretching, and proper backpack use.

*I have found it very useful because before I would carry my backpack down by my bum and I would struggle, and it was hell and suddenly you said I should raise it higher, and I thought to myself “it doesn’t weigh a thing! I was very surprised*.[19]

Teachers are concerned about the lack of attention given to the musculoskeletal effects of immobility, excessive technology use, dangerous play, and injury prevention. There is evidence of the effectiveness of SP programmes for this problem, but teachers do not have the training:

*The teachers aren’t taught about spinal health and what’s good for children and bad*.[29]

On the other hand, children and adolescents with spinal cord injuries face risks of spinal asymmetry, scoliosis, or hip dislocation, resulting in asymmetrical pressure distribution and an increased risk of pressure ulcers. There is a gap in prevention systems for these problems in schools. Paraplegic children are in need of training by physiotherapists for themselves and their peers, along with support from the school administration:

*[Assistance through physiotherapists] … there should be the likes of … the help of the hospital physiotherapist, and the school, and other special schools, we will manage*.[37]

#### 3.2.4. Implementing Alternative Therapies and Detecting Other Deficiencies

The inclusion of aquatic therapy programmes in schools and special education centres provides a fun and motivating form of physical activity. Physiotherapists report improvements in sensory experience, muscle tone and strength, motor spasticity, and breathing in children with cerebral palsy. As the physiotherapist explains:

*It is an environment where it is easier to mobilize compared to dry land conditions, so, first, you see what the person can do regarding mobility in the water, and then this can be applied in the classroom or the dining room, encouraging us to reinforce this*.[28]

Mindfulness techniques, breathing practices, body awareness, and yoga have all shown benefits for child development. Teachers and children report that the mindfulness and body awareness work, which is developed by physiotherapists, helps children concentrate. As one teacher sees it:

*For example, when doing homework, they can focus more; they used to chatter a lot. They can sit better and have a better posture. Like, in sorting out arguments and stuff, they already talk about it differently, and at some point, you gotta say, “F [referring to the physical therapist] comes with this and that…” [referring to the fact that she reminds them of what they’ve learned with the physical therapist, and it’s helped them to chill out]*.[35]

Older children with Developmental Coordination Disorder (DCD) or Autism Spectrum Disorder (ASD) are not considered to have motor disabilities but have unrecognised physiotherapy needs. This represents a gap that school physiotherapists must address:

*Children with DCD are not eligible for physiotherapy at school or child development centers after the age of 6*.[11]

### 3.3. The Challenge of Incorporating SP in Educational Settings

This topic covers the advantages and disadvantages, strengths and weaknesses of incorporating physiotherapists into specialised school settings.

#### 3.3.1. Integrating Therapy into Curriculum: Treating, and Supervising in Natural Settings

The availability of specialised staff facilitates physical activity for children with high support needs. Staff shortages limit opportunities for children to remain physically active. Physiotherapists are recognised as “physical activity experts” in the school setting. Parents comment that their children participate enthusiastically in activities when directed or suggested by physiotherapists:

*if she [physiotherapist] says ‘come on I want you to get up and go and walk around the school seven times’, [name] will go ‘okay, I’ll do it eight times for you’*.[33]

Observing adolescents in settings such as at home or in school is an essential part of physiotherapist interventions. One objective of SP is to facilitate the child’s participation in school by integrating therapy into their daily lives. When goals are integrated across different disciplines, progress is likely to be monitored and reported by several members of the educational team. As one physiotherapist explains:

*We all have something to say about that goal because it is within the context of that student’s day*.[3]

Another advantage is that physiotherapists can monitor the child over time, enabling the early detection of problems.

*By working in a school, physiotherapists have the opportunity to monitor the children’s progress over time and detect any signs of decline before they become more serious*.[11]

#### 3.3.2. Meeting Physiotherapy Needs in Special (And Regular) Education Schools

The school focuses on education; curricular demands prevent teachers from considering children’s other needs. Teachers feel they must focus on teaching and learning activities; they lack the time, capacity, or confidence to address the health needs of certain children.

*Our work has to be done, books have to be marked, assessments have to be done and marked and moderated and all of those so we don’t have time. We can’t worry about their spines because we are worried about what they’re learning*.[36]

SP could address these demands, but it is excluded from the department’s decisions. Nor is there a perceived political commitment to change this situation beyond special education. From this premise, the present study opens the challenge of incorporating SP into regular education.

*Implementing this model to mainstream schools means that physical therapists should be in the general (mainstream schools), and so it’s kind of new thought, so I think it’s going to be challenging*.[20]

Policymakers believe that some treatment for children with motor disabilities should be provided in special education settings; they are skeptical about its extension into regular education.

*Children with motor disabilities receive physiotherapy in special education schools. I don’t know whether to say that it is full, but certainly in special education schools there are physiotherapists who take care of these children… Those in mainstream education should also receive physiotherapy at school, but it doesn’t always happen. And if not in school, then where do they get it, I don’t know*.[11]

#### 3.3.3. Training Teachers and Other Professionals

For educational staff, it is important to attend physiotherapy sessions and practise the physiotherapist’s teachings. This way, they feel more motivated to dedicate time to certain techniques in class. For teachers, it would be advisable for the physiotherapist to train the teaching staff, consolidating what they have learned in class.

*And it is perfect that in parallel to this, F [the physical therapist] is also training us teachers, so I think it is an action, and the good thing is that it is a very global intervention and that it is not isolated and so on… Yes, so I think it is important. I also think it is important that we can introduce it, as I said, in a transversal way in the curriculum, that teachers are given the tools to be able to do this*.[35]

Physiotherapists may also supervise support staff who work with children with special needs in preschools. Support staff face difficulties when handling children to prevent fractures or injuries, assisting with position changes or transferring from a wheelchair to a standing device. Support staff identify training needs for managing children with cerebral palsy, and request the participation and involvement of school physiotherapists.

#### 3.3.4. Integrating into Multidisciplinary Teams to Overcome Challenges

The educational teams include children, parents, teachers, psychologists, occupational therapists, speech therapists, and physiotherapists. SP is committed to the integration and teamwork of all professionals. For educational team members, holistic care from multidisciplinary teams is the best option for children.

*We’ve all got the same goals. I obviously fight for education, physio fights for the physio, but I’m very mindful that there’s no point in just doing 100% education in school. You need to have some therapy as well*.[27]

When goals are integrated, the educational team gets involved. They monitor and report on the progress of intervention strategies. As one therapist comments:

*We work really hard to integrate goals … so that the teacher … or the staff can do it after we’re gone. So that they’re working on that same goal every day and not just when we’re there*.[3]

For school physiotherapists, knowing the opinions of students and parents is a key factor when it comes to involving them, setting goals, and achieving positive results in children’s treatment.

*I definitely want their [students’] input [on goal development] because … if they don’t buy into it, you’re not going to go anywhere with it anyway. I like them to come up with ideas of what problems they are having with their disability and … work with them on learning what their disability is … and how we can work on it*.[34]

Working with children and adolescents who are disabled and in pain is a challenging and exhausting experience. SPs not only apply techniques, but also establish a therapeutic relationship focused on improving quality of life. When there is not enough time to care for all of the children, they rely on the multidisciplinary team and trained assistants.

*I think you really have to collaborate with the other members of the team … we can’t be with a child all day every day. We don’t see the entire school day. We have lots of kids to serve … I think of specifically our kids who have a Physical Needs Assistant. That person is with them all day every day and is focused just on that one student*.[34]

For physiotherapists, achieving results is associated with setting goals with children, which makes them more independent and capable of making decisions.

*Some of the kids are bigger than I am and they get to the stage where they—if they’ve got knee flexion contractures, if it’s uncomfortable and they don’t want to do it, then they don’t do it*.[27]

Therapists mention the need for adaptations and environmental modifications in their interventions with students. This requires quality training, including school physiotherapy in students’ practical rotations. This quality training involves incorporating new skills and breaking out of the comfort zone that many veteran physiotherapists find themselves in. The inclusion of SPs in educational teams could involve a possible overlap of roles with other professionals. Developing specific skills in each area can help address this problem.

*There will be staff that will be threatened, there will be teachers that are threatened! There will be an occupational therapist that is threatened… it will be a real risk to implementation of this model if it’s not done correctly*.[20]

## 4. Discussion

The objective of this study was to identify qualitative evidence on the experiences and opinions of physiotherapists, children, parents, and teachers about school-based physiotherapy-led interventions. The results of this study highlight the need to integrate health promotion and therapy into the curriculum. Previous systematic reviews have shown positive results of physical activity on health and academic performance [40], but it is not always indicated for children with disabilities or special needs. SP specifically focuses on promoting children’s health and academic success [9,41]; a complex discipline that is not widely implemented due to the challenges of combining clinical and educational activities. While some studies describe models of SP provision in various countries [20], disparities in regulations, policies, and procedures lead to divergent practices [9,42]. Physiotherapists, teachers, and family members perceive the lack of communication and knowledge sharing among members of the educational team as a possible source of the problem. This situation generates a lack of scientific evidence and hinders the capacity of SP to integrate clinical skills within the educational setting [25,26]. This may be because in SP, evidence-based practice has been limited by a shortage of research and the complexity of decision-making [9]. Physiotherapists feel there is a lack of specific protocols and competencies in schools and that they cannot find a balance between education and therapy [27]. Although most clinical competencies in physiotherapy are relevant for practice in educational settings, the present study shows that criteria for practical application need to be unified. Participants perceive the need for multidisciplinary teams in educational centres [3,15,34], but efforts are focused mainly on education [26].

Although the literature corroborates the effectiveness of SP interventions in children’s motor development skills [24], some studies show a gap between ideal/current practice [22]. These differences underscore the need for deep reflection on the impact and empirical basis of SP interventions [25] and the institutional support for policy change [25,42]. This review highlights the positive perception of physiotherapists, family members, and teachers about SP interventions for children with specific conditions such as cerebral palsy [17], the use of standing frames [27], spinal pathology and back pain [19,29], spinal cord injury [37], aquatic therapy [28], aerobic exercise [17] or mind-body therapies [35]. Other studies suggest that SP could help prevent musculoskeletal disorders [13] and improve motor function, contributing to the academic and social development of children [43]. Families view SP as an opportunity for children’s psychomotor development [17,33,44]. They perceive children as more attentive and active, with improvements in self-concept [28]. Although the inclusion of SP within the educational team could generate controversies with physical education teachers [45], parents of children with disabilities support collaboration among professionals [46]. Other sources of controversy regarding the development of SP include a lack of information about which children require therapy, the cost-effectiveness of interventions [23], and confidence in their routine implementation in schools [7].

Teachers acknowledge the importance of physiotherapists in the inclusion of children with physical disabilities in the classroom [2,5], through teamwork in which occupational therapy and speech therapy converge [9]. Teachers, physiotherapists, and parents are committed to collaborating on the development of physical activities in schools [19,33]. SP could include teachers’ physical health promotion [14,18] and receiving training in posture, ergonomics, or using health technologies to support children with disabilities [27,36]. Overcoming these challenges implies building trust with the educational team [47], including physiotherapists, material, and specific training programmes in SP [10,11], along with integrating the interventions carried out in the school into the daily lives of the children [19,38].

## 5. Limitations

Conceptualizing a discipline as limited in scope as SP may have resulted in the omission of some relevant studies. Transferability of this metasynthesis may be limited, as school systems, curricula, and SP competencies are highly diverse. Cultural differences and varying levels of development in the countries where the studies were conducted may also hinder the formulation of broader conclusions and practical implications. Most of the authors included in this review were physiotherapists who have worked with children with special needs but have not performed the role of school physiotherapist within schools. Additionally, some of the original studies did not clearly specify the characteristics of their participants.

## 6. Rigour

Credibility: The authors reviewed the key concepts until reaching consensus during the screening, coding, and article analysis processes. Discrepancies were resolved by involving a third researcher. The original studies included various data collection methods. An intermediate meeting and a final meeting were held among the authors to approve the results. Transferability: the original studies described their research contexts in sufficient detail, allowing readers to assess whether their conclusions could be transferred to other settings. Confirmability: An audit trail was developed from the beginning of the study; however, no external researcher audit was conducted. Reflexivity: Most of the authors are educators in health sciences, which may have influenced the analysis of the original studies, including the extraction of codes, subthemes, and themes. The authors support the strengthening of SP and include individuals who either work in secondary education or focus on this topic in their doctoral research. Power dynamics among team members were respected, and consensus was reached at all stages of the research process. Contextually, this study forms part of the broader social expansion of Physiotherapy as an independent discipline within the health sciences in Spain.

## 7. Conclusions

Physiotherapists promote the establishment of collaborative goals aimed at strengthening the trust of children, families, and educational staff. They consider that SP addresses motor, communication, cognitive, and self-care challenges, adapting interventions to the time and resources available. They believe that SP needs to implement measurable objectives to generate scientific evidence and to take responsibility for advising parents and teachers in order to improve children’s well-being and safety. Physiotherapists may feel undervalued or isolated and perceive a lack of funding, time, and specific training to effectively implement SP. The children report perceived improvements from SP interventions in physical, communicative, leisure, and social participation domains. They view movement workshops, aquatic therapy, massage, breathing exercises, and posturology positively and consider them to be applicable at home. Some children see the physiotherapist as a key figure in meeting their needs and as someone who empowers their peers to provide support. Parents and family members endorse their children’s participation in SP activities and advocate for closer collaboration between physiotherapists and teachers. Parents feel that their voices are heard and perceive clear benefits of SP for their children, extending their learning beyond school to areas such as cerebral palsy and spinal health. Teachers also recognise the advantages of SP for children with disabilities, noting a shortage of physiotherapists to ensure the safe application of technologies in schools. They feel they lack the skills and training to address children’s motor difficulties and therefore call for the presence of physiotherapists in educational settings. According to teachers, SP can improve children’s physical and emotional well-being, posture, relaxation, and alertness. They also observe academic benefits and recommend that SP be included in the school curriculum.

## 8. Strengths and Implications

The present study calls for the development of shared objectives between health and education professionals. Children, parents, physiotherapists, and teachers perceive clear advantages in the implementation of SP within school settings. Participants consider that SP plays a vital role in supporting children with disabilities in schools, but there is a lack of evidence supporting its implementation and its impact on academic performance. The absence of clear guidelines and competencies hinders SP integration into educational teams.

## Figures and Tables

**Figure 1 healthcare-13-02859-f001:**
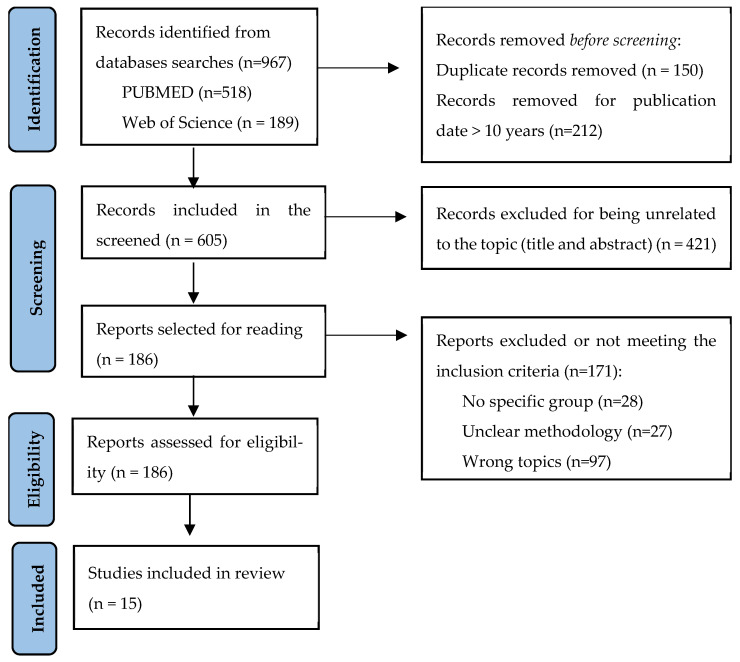
Flow chart.

**Table 1 healthcare-13-02859-t001:** Search strategy and results for each database.

Database	Search Strategy	Results
PubMed	((((school[Title/Abstract]) OR (education[Title/Abstract])) OR (physiotherapy[Title/Abstract])) AND (physical therapy[Title/Abstract])) AND (qualitative[Title/Abstract])	518
CINHAL	XB (physiotherapy) AND XB (school) AND XB (qualitative)	31
Web of Science	physiotherapy* (Topic) and school (Topic) and qualitative* (Topic)	189
SCOPUS	TITLE-ABS-KEY (physiotherapy* AND school AND qualitative)	229
	TOTAL:	967

This search was supplemented by a search of grey literature in Google Scholar, Doctoral Thesis Database (TESEO), and the Repository of the University of Almeria.

**Table 2 healthcare-13-02859-t002:** Quality assessment of studies (QARI) [32].

Article	1	2	3	4	5	6	7	8	9	10
Cleary et al. (2017) [17]	✔	✔	✔	✔	✔	✔	✔	✔	✔	✔
Cleary et al. (2019) [33]	✔	✔	✔	✔	✔	↔	↔	✔	✔	✔
Goodwin et al. (2019) [27]	✔	✔	✔	✔	✔	✘	✔	✔	✔	✔
Wynarczuk et al. (2019) [3]	✔	✔	✔	✔	✔	✔	✔	✔	↔	✔
Muñoz-Blanco et al. (2020) [28]	✔	✔	✔	✔	✔	✔	↔	✔	✔	✔
Blanco-Morales et al. (2020) [19]	✔	↔	✔	✔	✔	↔	↔	✔	✔	✔
Wynarczuk et al. (2020) [34]	✔	✔	✔	✔	✔	✔	↔	✔	✔	✔
Manamela et al. (2021) [26]	✔	✔	✔	↔	↔	✔	↔	✔	✔	✔
Louw et al. (2020) [29]	✔	↔	✔	✔	↔	✔	✔	✔	✔	✔
Cinar et al. (2022) [20]	✔	✔	✔	✔	✔	✔	↔	✔	✔	✔
López-Sierra et al. (2024) [35]	✔	✔	✔	✔	✔	✔	↔	✔	✔	✔
Fisher & Lown (2023) [36]	✔	✔	✔	✔	✔	✔	↔	✔	✔	✔
Rauter & Mathye (2024) [37]	✔	✔	✔	✔	✔	✔	↔	↔	✔	✔
Waiserberg et al. (2024) [11]	✔	✔	✔	✔	✔	✔	↔	✔	✔	✔
Kandal et al. (2025) [38]	✔	✔	✔	✔	✔	✔	✔	✔	✔	✔

✔ Yes, ↔ Unclear, ✘ No. 1. Congruence of philosophical perspective/methodology 2. Congruence of methodology/objectives 3. Congruence of methodology/data collection 4. Congruence of methodology/data analysis 5. Congruence of methodology/interpretation of results 6. Cultural and theoretical context of the researcher. 7. Influence of the researcher on the research 8. Participants represented. 9. Research Ethics Committee Approval 10. Conclusions from data analysis/interpretation.

**Table 3 healthcare-13-02859-t003:** Stages in the thematic synthesis process [39].

Stage	Description	Steps
STAGE 1	Text coding	Recall review question Read/re-read findings of the studies Line-by-line inductive coding Review of codes in relation to the text
STAGE 2	Development of descriptive themes	Search for similarities/differences between codes Inductive generation of new codes Write preliminary and final report
STAGE 3	Development of analytical themes	Inductive analysis of sub-themes Individual/independent analysis Pooling and group review

**Table 4 healthcare-13-02859-t004:** Characteristics of selected studies.

Author and Year	Country	Sample	Design	Data Collection	Data Analysis	Main Theme
Cleary et al. (2017) [17]	Australia	Childs (8), parents (7), teachers (6), PT (7)	Qualitative descriptive study	SSI	Thematic analysis	Participants recognise benefits of an aerobic exercise programme for children with cerebral palsy
Cleary et al. (2019) [33]	Australia	Child (10), parents (13), teachers (27), PT (23)	Qualitative descriptive study	FGs	Thematic analysis	Identifying barriers/facilitators of physical activity for young people in specialised schools
Goodwin et al. (2019) [27]	UK	PT (9), teachers (8), parents (9), Mixed (17)	Qualitative study	FGs	Framework method	Training and transdisciplinary communication are required to balance therapy versus education
Wynarczuk et al. (2019) [3]	USA	PT (20)	Qualitative descriptive study	FGs	Thematic analysis	PT must understand individualised goals, influence services, and optimise student outcomes
Muñoz-Blanco et al. (2020) [28]	Spain	Child (14), parents (8), PT (5)	Qualitative case study with embedded units	Non-participant observations, IDI, FGs	Thematic analysis	Participants perceive aquatic therapy as an alternative treatment approach that can be applied in schools
Blanco-Morales et al. (2020) [19]	Spain	Child (49), teachers (9), parents (11), PT (9)	Collaborative action research	IDI, FGs, reflexive diaries, field notes	Inductive analysis	Scholar physiotherapy programmes offer students new tools to decrease their back pain
Wynarczuk et al. (2020) [34]	USA	PT (20)	Qualitative descriptive study	FGs	Thematic analysis	SP professionals recommend working collaboratively with students, parents, and members of the educational team to achieve shared goals
Manamela et al. (2021) [26]	South Africa	PT (22)	Mixed method research	FGs	Thematic analysis	Perception of physiotherapists’ role in inclusive education
Louw et al. (2021) [29]	South Africa	Child (43), parents (17), teachers (33)	Qualitative descriptive study	IDIs, FGs	Inductive analysis	There is a need for further engagement on school-based spinal health promotion programmes
Cinar et al. (2022) [20]	Different countries (8)	PT (38)	Qualitative study	FGs	Framework method	Perceived strengths and weaknesses of a collaborative tiered school-based physiotherapy (PT) service delivery model
López-Sierra et al. (2024) [35]	Spain	Child (43), teachers (2)	Qualitative descriptive study	IDIs, FGs	Thematic analysis	Children’s and teachers’ perceptions of participation in a Mind–Body Activity Programme led by a PT
Fisher & Lown (2023) [36]	South Africa	Teachers (37)	Qualitative descriptive study	IDIs, FGs	Inductive analysis	Teachers perceive a lack of evidence regarding the implementation of movement strategies in the classroom
Rauter & Mathye (2024) [37]	South Africa	Child (12)	Qualitative descriptive study	IDIs, FGs	Thematic analysis	The school physiotherapist’s role in peer support in preventing pressure ulcers in students with paraplegia in special schools
Waiserberg et al. (2024) [11]	Israel	PT (10)	Qualitative descriptive study	IDIs	Inductive analysis	Policymakers question the provision of physiotherapy services in schools
Kandal et al. (2025) [38]	Norway	PT (13)	Qualitative descriptive study	FGs	Thematic analysis	Physiotherapists emphasise the need to integrate their interventions into the daily lives of adolescents with pain, including in schools

PT = Physiotherapist. IDI = In-Depth Interview. FGs = Focus Groups. SSI = Semi-Structured Interview.

**Table 5 healthcare-13-02859-t005:** Themes and subthemes.

Themes	Subthemes	Unit of Meaning
3.1 Clinical and school physiotherapy: separated by a fine line	3.1.1 Lack of specific guidelines and ambiguity of professional competencies	Lack of guidelines, organising practice, differences between countries, ambiguity of competencies, lack of accountability, shared responsibility
	3.1.2 Lack of support: feeling like outsiders	Lack of managerial support, learning by doing, decision-making, lack of laterals and personnel, undervalued in the educational team
	3.1.3 Possible professional intrusion	Possible professional intrusion, physiotherapy, and sports
	3.1.4 Children-centred, evidence-based	Evidence-based physiotherapy, motor skills, independence of movement, measurable objectives
3.2. Ensuring healthcare for children with specific conditions in schools	3.2.1 Care for children with cerebral palsy	Aerobic exercise programme, cerebral palsy, motor, and psychological benefits
	3.2.2 Supervising children using standing frames	Training needs, complications, burns, train teachers
	3.2.3 Prevention and treatment of back problems/pain and complications from spinal cord injuries	Ergonomics, stretching, sedentary lifestyle, class backpack, movement, friction, ulcer prevention
	3.2.4 Implementing alternative therapies and detecting other deficiencies	Aquatic therapy, mindfulness, breathing, body awareness, yoga, minor motor limitations
3.3 The challenge of incorporating SP in educational settings	3.3.1 Integrating therapy into curriculum: treating, and supervising in natural settings	Daily observation, early identification of problems, professional recognition, familiar recognition, child recognition
	3.3.2 Meeting physiotherapy needs in special (and regular) education schools	Time, skills, funding, security, severity, doubts
	3.3.3 Training teachers and other professionals	Training teachers and assistants, learning from the physiotherapist
	3.3.4 Integrating into multidisciplinary teams to overcome challenges	Educational team, common goals, Technology inclusion in the classroom, lack of time, training and qualified staff

**Table 6 healthcare-13-02859-t006:** Contribution matrix of studies to themes/subthemes.

Themes	Studies Contribution
**3.1 Clinical and school physiotherapy: separated by a fine line**	
3.1.1 Lack of specific guidelines and ambiguity of professional competencies	[11,20,27,34]
3.1.2 Lack of support: feeling like outsiders	[20,26,33,34,36]
3.1.3 Possible professional intrusion	[11,33,36]
3.1.4 Children-centred, evidence-based	[19,27,28,29,34]
**3.2. Ensuring healthcare for children with specific conditions in schools**	
3.2.1 Care for children with cerebral palsy	[17,27,28,33]
3.2.2 Supervising children using standing frames	[27,28]
3.2.3 Prevention and treatment of back problems/pain and complications from spinal cord injuries	[19,29,36,37]
3.2.4 Implementing alternative therapies and detecting other deficiencies	[11,28,35]
**3.3 The challenge of incorporating SP in educational settings**	
3.3.1 Integrating therapy into curriculum: treating, and supervising in natural settings	[11,19,26,27,29,33,34,36,37]
3.3.2 Meeting physiotherapy needs in special (and regular) education schools	[11,20,27,29,33,34,36]
3.3.3 Training teachers and other professionals	[3,26,27,29,33,35]
3.3.4 Integrating into multidisciplinary teams to overcome challenges	[3,19,20,26,27,29,33,34]

**Table 7 healthcare-13-02859-t007:** Coding example: from code to theme.

Themes	Subthemes	Unit of Meaning	Code	Quotes
3.2. Ensuring healthcare for children with specific conditions in schools	3.2.1 Care for children with cerebral palsy		Parent and student satisfaction	*…I’m grateful for that*. (Parent) [17] *Happy! It makes me feel good*. (Student) [17]
			Developing independence	*kids need to be active. that’s the way that they learn* (Teacher) [33] *“Their posture is better when they leave and as a result of this better posture they are more willing to do things for their hygiene or clothing.* (Therapist) [28]
		Motor, psychological and social benefits	Positive social experience	*A few people noticed that she was becoming more social in the classroom…* (Teacher) [17]
		Balancing education and therapy	Recognizing the work of the physiotherapist	*if she [physiotherapist] says ‘come on I want you to get up* *and go and walk around the school seven times’, [name] will go ‘okay, I’ll do it eight times for you’.* (Parent) [33] *… I’m very mindful that there’s no point just doing 100% education in school. You need to have some therapy as well.* (Education group) [27]
		Overcoming barriers	Drowsiness, fatigue, time and material limitations	*If they have adjusted it and if you say you are going to get one for one of the kids, someone else might have been in and you have got to adjust every time-it just takes up loads of time.* [Education group] [27]

## Data Availability

No new data were created or analyzed in this study. Data sharing is not applicable to this article.

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
