# Peer review of "Experiences and Opinions of Physiotherapists, Children, Families, and Teachers About School-Based Physiotherapy-Led Interventions: A Metasynthesis of Qualitative Studies"

_healthcare, 2025, doi:10.3390/healthcare13222859_

Round 1
Reviewer 1 Report
Comments and Suggestions for Authors
Abstract
The objectives currently refer to the views of several groups, but the results and conclusions do not clarify whether the three key themes originate from children, professionals, or the literature.
It is therefore recommended that the objectives and conclusions are aligned and that overly bold or vague statements are avoided.
Objectives should be precise and reproducible, enabling readers to understand exactly what is to be investigated.
Introduction
Lines 38–47: Some of the references are outdated and could be replaced or supplemented with more recent versions (suggested references: 2–6).
Lines 48–50: The indirect mention of the WCPT duplicates information from reference 1. It is therefore recommended that references 7 and 8 are removed.
Figure 1 would benefit from including an overview of the reasons for excluding studies by title or abstract, making the selection process more transparent.
It is necessary to clarify whether any quality assessment criteria were applied to the included studies (e.g. percentage of JBI items), and whether all studies that met the inclusion criteria were included.
Separator lines should be inserted at Table 3 between studies to make the table easier to read.
Add more information to the final column so that readers do not need to refer to the original studies to understand the presented data.
For example, explain how the study 'Benefits of an aerobic exercise programme for children with cerebral palsy' relates to the opinion on school physiotherapy.
Correct any omissions, such as the incomplete sentence in the study by Blanco-Morales et al. (2020): 'Physiotherapy activities offer students new tools to decrease their back pain and improve their health' (close the parentheses).
Table 4 is very clear. I think you could use subtopics to further develop the summary. Only now can we understand the three major themes and how they were identified.
Discussion: Address the transdisciplinary approach by demonstrating how physical therapists equip caregivers, teachers, and assistants with the necessary skills to support children at school.
Focus on studies that directly address this interaction.
Conclusion:
It is recommended that the objectives are clarified or the conclusion is rewritten to align with what was actually investigated in the study.
Comments on the Quality of English LanguageA general review of the English is suggested to ensure greater formality and consistency. Personal terms such as 'our study' should be avoided; indirect forms such as 'in the present study' should be used throughout the text instead.
Author Response
Response to reviewers
REVIEWER 1
The authors would like to thank Reviewer 1 for their contributions to improving the study.
|
QUESTIONS |
RESPONSE |
|
Abstract
The objectives currently refer to the views of several groups, but the results and conclusions do not clarify whether the three key themes originate from children, professionals, or the literature.
It is therefore recommended that the objectives and conclusions are aligned and that overly bold or vague statements are avoided.
Objectives should be precise and reproducible, enabling readers to understand exactly what is to be investigated.
|
The title of the study has been changed, per the recommendations of all the reviewers.
TITLE: Experiences/opinions of physiotherapists, children, families, and teachers about school-based physiotherapy-led interventions: A metasynthesis of qualitative studies.
We have included the experiences/opinions of physiotherapists, children, parents, and teachers concerning SP services. We excluded studies about the general population and non-experience studies. The Sørvoll et al. (2018) and Walton (2020) articles have been removed from the selection after re-reading the objectives and the conclusions.
The results have been reviewed, including experiences from all groups studied. The conclusions have been reviewed to align them with the objectives and have been rewritten, differentiating between groups of participants.
|
|
Introduction |
|
|
Lines 38–47: Some of the references are outdated and could be replaced or supplemented with more recent versions (suggested references: 2–6).
|
The references indicated by the reviewer have been revised and more recent and updated references have been added. |
|
Lines 48–50: The indirect mention of the WCPT duplicates information from reference 1. It is therefore recommended that references 7 and 8 are removed. |
References 7 and 8 have been removed from this part of the text in order to avoid duplicate information about the WCPT. |
|
Figure 1 would benefit from including an overview of the reasons for excluding studies by title or abstract, making the selection process more transparent. |
Reasons for exclusion: after reading the title and abstract, the article does not clearly respond to the objectives of the study. Figure 1 has been modified (p.3) |
|
It is necessary to clarify whether any quality assessment criteria were applied to the included studies (e.g. percentage of JBI items), and whether all studies that met the inclusion criteria were included. |
The review includes all the studies that fulfilled ≥ 70% of the QARI quality criteria. No studies were excluded after quality assessment (p.4). |
|
Separator lines should be inserted at Table 3 between studies to make the table easier to read.
|
Separator lines have been inserted in Table 3 between studies. |
|
Add more information to the final column so that readers do not need to refer to the original studies to understand the presented data. |
Information in the final column of Table 3 has been modified and completed. |
|
For example, explain how the study 'Benefits of an aerobic exercise programme for children with cerebral palsy' relates to the opinion on school physiotherapy. |
The study explores perceptions of physiotherapists and teachers regarding the effects of an aerobic exercise program taught in schools for young people with cerebral palsy. School physiotherapists note improvements in the children's motor, communication, and relationship skills. Teachers note advantages to children's participation in an SP (school physiotherapy) program. |
|
Correct any omissions, such as the incomplete sentence in the study by Blanco-Morales et al. (2020): 'Physiotherapy activities offer students new tools to decrease their back pain and improve their health' (close the parentheses). |
The parentheses has been closed in Table 3, Blanco-Morales et al. (2020). |
|
Table 4 is very clear. I think you could use subtopics to further develop the summary. Only now can we understand the three major themes and how they were identified.
|
We thank the reviewer for their comment. However, Table 4 is already structured into 3 themes and 12 subthemes. We believe this division is sufficient to capture the experiences and opinions of all participants. |
|
Discussion: Address the transdisciplinary approach by demonstrating how physical therapists equip caregivers, teachers, and assistants with the necessary skills to support children at school. |
The discussion section has been largely rewritten, highlighting participants' opinions and experiences regarding the strengths of SP implementation and the support provided by physical therapists to children, families, and teachers. |
|
Focus on studies that directly address this interaction. |
The changes made throughout the article aim to address this interaction. |
|
Conclusion: It is recommended that the objectives are clarified or the conclusion is rewritten to align with what was actually investigated in the study.
|
The conclusion has been rewritten to align it with the study objectives and the groups of participants who recount their experiences. |
|
Comments on the Quality of English Language A general review of the English is suggested to ensure greater formality and consistency. Personal terms such as 'our study' should be avoided; indirect forms such as 'in the present study' should be used throughout the text instead. |
The quality of the English language has been reviewed. Personal terms have been removed. |
Reviewer 2 Report
Comments and Suggestions for Authors
Dear Authors,
Please, see few comments you might consider or clarify
Title and aims state a metasynthesis of experiences/opinions regarding school physiotherapy (SP), focused on children with disabilities/special motor needs. However, the included set mixes:
school programs for general student populations (e.g., back-pain/spinal health workshops, classroom movement) and mind–body activities not clearly delivered under SP;
studies whose primary focus is not experiences/opinions (e.g., program evaluations).
Option A (narrow): Experiences/opinions about SP for children with disabilities/special motor needs. Re-screen and exclude general-population and non-experience studies.
Option B (broaden): Experiences/opinions about any school-based physiotherapy-led interventions. Retitle, revise background & inclusion criteria, and justify inclusion of general populations.
Specify platforms (e.g., PubMed interface, WoS Core Collection timespan), dates searched, and who conducted each search.
List grey sources searched (e.g., OpenGrey, ProQuest Dissertations, policy repositories) and how screened.
Provide PRISMA-S checklist in Supplement.
PRISMA flow shows counts that are hard to reconcile (e.g., “Reports excluded (n=111); incorrect outcomes (n=40); incorrect population (n=18)” summing to 169 yet labels are ambiguous).
Screening in duplicate is not clearly stated for title/abstract and full text; only data extraction mentions “independently”.
Confirm dual-reviewer screening at each stage with a tie-breaker; report agreement handling.
Provide a reasons-for-exclusion table for all full-text exclusions (study-level).
Ensure numbers sum correctly and match all stages.
Inclusion says qualitative or mixed-methods on experiences/opinions; several included studies appear primarily interventional/quantitative or policy/description, and some concern non-SP implementers.
Map each included study explicitly to SPIDER (Sample, Phenomenon, Design, Evaluation, Research type) showing it truly meets the phenomenon of interest (SP experiences/opinions).
Re-screen and exclude studies that do not meet the phenomenon or design requirements, or revise the phenomenon to match what was actually included.
JBI checklist results are summarized with ✔/↔/✘, but you then state all studies are “high quality,” which contradicts the table and offers no rationale.
No sensitivity analysis by quality, and no confidence in findings.
Coding is described in generalities; there is no codebook, no exemplar quotes table per subtheme, and no study-by-theme contribution matrix.
Some quotations are context-light or feel tangential; traceability to sources is unclear.
Provide an audit trail: initial codes → descriptive themes → analytical themes, with illustrative quotes (source, page/line if available).
Clarify reflexivity/positionality (how authors’ backgrounds and any prior work in SP might shape interpretation).
Disclose self-citation/inclusion of authors’ primary studies and explain mitigations (e.g., independent screening/extraction by non-overlapping team member).
Conclusions claim a “lack of evidence supporting implementation and impact on academic performance,” but the review did not synthesize academic outcomes; it synthesized experiences/opinions.
Several included items (e.g., manipulative therapy RCTs, general back-pain classes) are not clearly SP and risk over-generalization.
Reframe all conclusions to align strictly with qualitative evidence on experiences/opinions.
Where you discuss outcomes (academic or motor), cite separate quantitative syntheses as context, not as findings of this review.
Avoid normative claims about “professional intrusion” unless grounded in repeated qualitative evidence and balanced with the roles of OT/PE staff.
Best wishes
Author Response
REVIEWER 2
The authors would like to thank Reviewer 2 for their contributions to improving the study.
|
QUESTIONS |
RESPONSE |
|
Dear Authors,
Please, see few comments you might consider or clarify
Title and aims state a metasynthesis of experiences/opinions regarding school physiotherapy (SP), focused on children with disabilities/special motor needs. However, the included set mixes:
school programs for general student populations (e.g., back-pain/spinal health workshops, classroom movement) and mind–body activities not clearly delivered under SP;
studies whose primary focus is not experiences/opinions (e.g., program evaluations).
Option A (narrow): Experiences/opinions about SP for children with disabilities/special motor needs. Re-screen and exclude general-population and non-experience studies.
Option B (broaden): Experiences/opinions about any school-based physiotherapy-led interventions. Retitle, revise background & inclusion criteria, and justify inclusion of general populations. |
The title of the study has been changed, per the recommendations of all the reviewers.
TITLE: Experiences/opinions of physiotherapists, children, families, and teachers about school-based physiotherapy-led interventions: A metasynthesis of qualitative studies.
We have included experiences/opinions of physiotherapists, children, parents, and teachers concerning SP services in the study.
We excluded general-population and non-experience studies. Two studies were withdrawn from the selection after rereading the objectives and full text.
● Sørvoll et al., 2018 (Essentially includes experiences of PT-aides). ● Walton 2020 (focuses on physiotherapists' perspectives on the future of their profession, not on school physiotherapy)
The rest of the studies have been maintained in order to gather experiences from physiotherapists, children, families, and teachers in the school environment. The results and conclusions have been reviewed, to align them with the objectives and the groups whose experiences have been described. |
|
Specify platforms (e.g., PubMed interface, WoS Core Collection timespan), dates searched, and who conducted each search. |
Information concerning the databases consulted, time periods, and which researchers conducted the search have been completed (pp. 2-3). |
|
List grey sources searched (e.g., OpenGrey, ProQuest Dissertations, policy repositories) and how screened. |
We have added a list of sources of grey literature consulted, such as Google Scholar, the University of Almería Repository, and the Doctoral Thesis Database (TESEO). |
|
Provide PRISMA-S checklist in Supplement. |
The PRISMA-S checklist has been added in Supplement. In view of the doubts raised by reviewer 3, the ENTREQ declaration checklist has also been added. |
|
PRISMA flow shows counts that are hard to reconcile (e.g., “Reports excluded (n=111); incorrect outcomes (n=40); incorrect population (n=18)” summing to 169 yet labels are ambiguous).
|
Figure 1 has been changed and rewritten to conform to the 2020 Flow Chart Model. Search data has been supplemented and clarified (there was an error in the previous version). We have provided reasons for exclusion of articles after reading the full text. Two articles have been excluded, bringing the total number of articles included in the review to 15. |
|
Screening in duplicate is not clearly stated for title/abstract and full text; only data extraction mentions “independently”. |
The elimination of duplicates due to matches in title/abstract and full text has been specified in the Flow Chart (Figure 1). |
|
Confirm dual-reviewer screening at each stage with a tie-breaker; report agreement handling.
|
Dual-reviewer screening in each stage has been explained in sections 2.4 and 2.5, as well as how agreements were reached for the inclusion of studies in each phase. |
|
Provide a reasons-for-exclusion table for all full-text exclusions (study-level).
|
Reasons for exclusion of articles after reading the full text have been added to the Flow chart, p. 3. |
|
Ensure numbers sum correctly and match all stages.. |
The sum of the numbers has been checked (Flow chart, p. 3). |
|
Inclusion says qualitative or mixed-methods on experiences/opinions; several included studies appear primarily interventional/quantitative or policy/description, and some concern non-SP implementers.
|
We have checked and confirmed the inclusion criteria for qualitative and mixed studies. References to studies incorporating interventions focus on the participants’ experiences in those interventions. Articles related to policies or legal provisions include the opinions of participants. Two studies were removed from the selection after rereading the objectives and full text. • Sørvoll et al., 2018 (Essentially includes experiences of PT-aides). • Walton 2020 (focused on the perspectives of physiotherapists on the future of their profession, not on school physiotherapy). |
|
Map each included study explicitly to SPIDER (Sample, Phenomenon, Design, Evaluation, Research type) showing it truly meets the phenomenon of interest (SP experiences/opinions). |
In the authors' opinion, Table 3 meets the reviewer's requirements. The 15 studies included comply with the SPIDER method (p. 2). |
|
Re-screen and exclude studies that do not meet the phenomenon or design requirements, or revise the phenomenon to match what was actually included.
|
The study search and selection process has been re-examined. Two studies have been removed: • Sørvoll et al., 2018 (Essentially includes experiences of PT-aides). • Walton 2020 (focused on the perspectives of physiotherapists on the future of their profession, not on school physiotherapy).
|
|
JBI checklist results are summarized with ✔/↔/✘, but you then state all studies are “high quality,” which contradicts the table and offers no rationale. |
The following clarification has been added (p. 4): The review includes all studies that met ≤ 70% of the QARI criteria. |
|
No sensitivity analysis by quality, and no confidence in findings.
|
Sensitivity analysis studies the individual influence of each study on the final result of a meta-analysis. The authors understand that this requirement is more appropriate for non-qualitative studies (any biases in different sections of selected studies have been analysed using the QUARI tool). |
|
Coding is described in generalities; there is no codebook, no exemplar quotes table per subtheme, and no study-by-theme contribution matrix. |
We have included (p. 7) a matrix of each study’s contribution to the topics/subtopics (Table 5). |
|
Some quotations are context-light or feel tangential; traceability to sources is unclear.
|
The quotations have been revised to fit the context of the original study. Several quotations have been removed and replaced with others. |
|
Provide an audit trail: initial codes → descriptive themes → analytical themes, with illustrative quotes (source, page/line if available). |
An example of an inductive data analysis process from code to Theme has been included (p. 8). |
|
Clarify reflexivity/positionality (how authors’ backgrounds and any prior work in SP might shape interpretation). |
We have added a section on rigour, including reflexivity. |
|
Disclose self-citation/inclusion of authors’ primary studies and explain mitigations (e.g., independent screening/extraction by non-overlapping team member). |
A description of the data selection and extraction process has been improved and lengthened. |
|
Conclusions claim a “lack of evidence supporting implementation and impact on academic performance,” but the review did not synthesize academic outcomes; it synthesized experiences/opinions. |
The study conclusions have been rewritten to align them with the objectives and participants’ opinions and experiences. |
|
Several included items (e.g., manipulative therapy RCTs, general back-pain classes) are not clearly SP and risk over-generalization.
|
The study by Dissing et al. (2018) is an RCT focusing on recurrent back pain in children aged 9 to 15 years in a school setting after physiotherapy. Based on the reviewer's instructions, it has been removed from the discussion. |
|
Reframe all conclusions to align strictly with qualitative evidence on experiences/opinions
|
The study conclusions have been rewritten to better align them with the objectives and participants’ opinions and experiences.
|
|
Where you discuss outcomes (academic or motor), cite separate quantitative syntheses as context, not as findings of this review. |
We have followed the reviewer’s recommendation throughout the text. |
|
Avoid normative claims about “professional intrusion” unless grounded in repeated qualitative evidence and balanced with the roles of OT/PE staff. |
Per the reviewer’s recommendation, we have made the following changes: ● “offer” for “could offer” (p. 9 ). ● The name of Subtheme 3.1.4 has been changed, and it now reads: Possible professional intrusion. ● Text: “professional intrusion” has been changed to “possible professional intrusion”. |
Reviewer 3 Report
Comments and Suggestions for Authors
-
High similarity index (24%): The similarity level is elevated and may indicate issues with textual overlap or insufficient paraphrasing in several sections. This should be carefully reviewed to avoid concerns regarding originality.
-
Title: The title could be made more concise and informative to reflect the study design and target population more precisely.
-
Abstract:
-
Excessive methodological detail.
-
Lack of clear description of implications and novelty.
-
Overlap between objectives and conclusions.
-
-
Introduction:
-
Overly extensive and partly descriptive rather than critical.
-
Repetition of background information already stated later.
-
Limited justification of the research gap and scientific contribution.
-
-
Methods:
-
Search strategy insufficiently detailed regarding Boolean structure and filters.
-
Lack of registration statement (e.g., PROSPERO or equivalent).
-
Inconsistent use of the SPIDER and PRISMA frameworks.
-
The quality assessment (QARI) table includes unclear coding and lacks full transparency on reviewer agreement.
-
Data extraction process not sufficiently reproducible.
-
-
Results:
-
Excessive narrative length and redundancy between tables and text.
-
Some subthemes overlap conceptually, reducing clarity of synthesis.
-
Lack of quotations in certain thematic areas to support interpretation.
-
-
Figures and Tables:
-
Flowchart does not follow updated PRISMA 2020 format.
-
Table captions are incomplete and inconsistent in style.
-
Tables 1–3 need clearer legends and source indications.
-
-
Discussion:
-
Reiterates results rather than interpreting them.
-
Insufficient linkage with existing literature beyond descriptive comparison.
-
Does not highlight methodological strengths or potential transferability.
-
Overgeneralization of qualitative findings without reflexivity.
-
-
Limitations:
-
Superficial; lacks detail on methodological and analytical constraints.
-
-
Conclusions:
-
Too similar to abstract; limited originality and synthesis.
-
No explicit indication of implications for clinical or educational policy.
-
References:
-
Inconsistent formatting with MDPI style.
-
Several entries show outdated URLs or missing DOIs.
-
Reference range (2015–2025) includes future-dated articles (2025), which requires verification.
-
Language and Style:
-
Frequent lexical repetition and inconsistent tense use.
-
Occasional direct translation structures that reduce fluency.
-
Overuse of passive voice in the results and discussion.
-
Overall structure:
-
Imbalance between extensive results and brief methodological justification.
-
Lack of a distinct “Strengths and implications” section.
-
Overlap between “Discussion” and “Conclusion” sections.
Author Response
REVIEWER 3
The authors would like to thank Reviewer 3 for their contributions to improving the study.
|
High similarity index (24%): The similarity level is elevated and may indicate issues with textual overlap or insufficient paraphrasing in several sections. This should be carefully reviewed to avoid concerns regarding originality.
|
The similarity index of the article has decreased following this revision. High values correspond to the incorporation of methods, quotations, or conclusions taken from the original studies. The authors do not understand the reviewer's doubts regarding the originality of the article. |
|
Title: The title could be made more concise and informative to reflect the study design and target population more precisely.
|
We thank the reviewer for this very true comment. We have changed the title of the study, incorporating the recommendations of all reviewers, so that it more accurately reflects the design and target population of the study:
TITLE: Experiences/opinions of physiotherapists, children, families, and teachers about school-based physiotherapy-led interventions: A metasynthesis of qualitative studies.
We included experiences/opinions of physiotherapists, children, parents, and teachers about SP services in the study.
Studies of the general population and studies without experience were excluded from this review.
|
|
Abstract:
Excessive methodological detail.
Lack of clear description of implications and novelty.
Overlap between objectives and conclusions. |
Some of the methodological details in the abstract have been removed. A sentence has been added about the implications of the study: Understanding the experiences of physiotherapists, children, parents and teachers may help improve SP practice. The objectives of the study have been adjusted and the conclusions rewritten to avoid overlap. |
|
Introduction:
Overly extensive and partly descriptive rather than critical.
Repetition of background information already stated later.
Limited justification of the research gap and scientific contribution.
|
The introduction to the study has been largely rewritten. It now flows better, is less lengthy and descriptive, and any repetition has been removed. References 7 and 8 have been deleted so as not to duplicate information on WCPT, and will be incorporated later in the text.
The rationale for the study is the lack of up-to-date review studies on the experiences of physiotherapists, children, family members and teachers regarding school-based physiotherapy-led interventions. |
|
Methods:
Search strategy insufficiently detailed regarding Boolean structure and filters.
Lack of registration statement (e.g., PROSPERO or equivalent).
Inconsistent use of the SPIDER and PRISMA frameworks.
The quality assessment (QARI) table includes unclear coding and lacks full transparency on reviewer agreement.
Data extraction process not sufficiently reproducible.
|
The search strategy used is specified in greater detail (see Table 1).
This review has not been included in Prospero, so we cannot provide a registration statement.
The study followed the SPIDER framework to generate the research question. This is specified in the text (p. 2-3). All studies included fulfill the criteria: S (sample) fisioterapeutas, niños, padres, maestros. PI (Phenomenon of Interest): school-based physiotherapy-led interventions. D (Design): qualitative studies (qualitative descriptive study, Phenomenological study, ..). E (Evaluation): experiences and opinions. R (Research): qualitative study, mixed methods study.
In response to reviewers' concerns about inconsistent use of the PRISMA-S Declaration (systematic reviews), which is more appropriate for meta-analyses of quantitative studies, the authors have also chosen to incorporate the Enhancing Transparency in Reporting the Synthesis of Qualitative Research (ENTREQ) declaration (more in line with meta-synthesis articles).
The quality assessment table (QARI) has been revised. An explanation regarding data extraction and reviewer agreement has been added (p. 3).
The search strategy and data extraction process have been specified in greater detail (see Table 1 and Figure 1).
|
|
Results:
Excessive narrative length and redundancy between tables and text.
Some subthemes overlap conceptually, reducing clarity of synthesis.
Lack of quotations in certain thematic areas to support interpretation.
|
The summary of results at the beginning of the results section has been restructured (two articles have been removed from the review).
Some subtopics have been combined so that they do not overlap conceptually. There are now three topics and twelve subtopics.
Quotations from removed articles have been deleted.
Quotations supporting some subtopics have been included.
|
|
Figures and Tables:
Flowchart does not follow updated PRISMA 2020 format.
Table captions are incomplete and inconsistent in style.
Tables 1–3 need clearer legends and source indications. |
The flow chart now follows the updated Prisma 2020 format.
Table titles and captions have been revised and completed, indicating sources.
Titles of all tables revised. Authors do not understand what the reviewer means by ‘inconsistent in style’. |
|
Discussion:
Reiterates results rather than interpreting them.
Insufficient linkage with existing literature beyond descriptive comparison.
Does not highlight methodological strengths or potential transferability.
Overgeneralization of qualitative findings without reflexivity. |
The discussion section has been largely rewritten, highlighting participants' opinions and experiences regarding the strengths of SP implementation and the support provided by physiotherapists to children, families, and teachers. The interpretation of results and links to the literature have been reinforced.
References to the transferability of the results have been incorporated into the Rigor section (p. 15).
Reflexivity has been included in the Rigor section (p. 15). |
|
Limitations:
Superficial; lacks detail on methodological and analytical constraints. |
The limitations section has been completed and rewritten. (p. 15). |
|
Conclusions:
Too similar to abstract; limited originality and synthesis.
No explicit indication of implications for clinical or educational policy. |
The study conclusions are rewritten to align them with the objectives and participants’ opinions and experiences.
A section on strengths and implications has been included at the end of the study. |
|
References:
Inconsistent formatting with MDPI style.
Several entries show outdated URLs or missing DOIs.
Reference range (2015–2025) includes future-dated articles (2025), which requires verification. |
The References section has been completely revised to update URLs and adapt to the MPDI style.
The search has now been extended to October 2025. |
|
Language and Style:
Frequent lexical repetition and inconsistent tense use.
Occasional direct translation structures that reduce fluency.
Overuse of passive voice in the results and discussion. |
We have revised the language and style, removing lexical repetition and checking verb tenses.
|
|
Overall structure:
Imbalance between extensive results and brief methodological justification. Lack of a distinct “Strengths and implications” section.
Overlap between “Discussion” and “Conclusion” sections.
|
The imbalance between extensive results and brief methodological justification has been partially corrected. The results are comprehensive and derived from the original studies. The justification for the study is, in essence, the absence of qualitative syntheses on this topic.
We have included a separate section on ‘Strengths and implications’.
The study's conclusions are rewritten to align them with the objectives, participants’ opinions and experiences, and to avoid overlap with the discussion.
|
Round 2
Reviewer 2 Report
Comments and Suggestions for Authors
Thank you so much for addressing the comments
Reviewer 3 Report
Comments and Suggestions for Authors
The authors have correctly integrated the requested changes, and the manuscript is accepted in its present form.